# An atlas of brain-bone sympathetic neural circuits in mice

Vitaly Ryu[1,2]*, Anisa Azatovna Gumerova[1,2], Ronit Witztum[1,2], Funda Korkmaz[1,2], Liam Cullen[1,2], Hasni Kannangara[1,2], Ofer Moldavski[1,2], Orly Barak[1,2], Daria Lizneva[1,2], Ki A Goosens[1,3], Sarah Stanley[2], Se-Min Kim[1,2], Tony Yuen[1,2], Mone Zaidi[1,2]*

[1]Center for Translational Medicine and Pharmacology, Icahn School of Medicine at Mount Sinai, New York, United States; [2]Department of Medicine and of Pharmacological Sciences, Icahn School of Medicine at Mount Sinai, New York, United States; [3]Department of Psychiatry, Icahn School of Medicine at Mount Sinai, New York, United States

**Abstract** There is clear evidence that the sympathetic nervous system (SNS) mediates bone metabolism. Histological studies show abundant SNS innervation of the periosteum and bone marrow–these nerves consist of noradrenergic fibers that immunostain for tyrosine hydroxylase, dopamine beta-hydroxylase, or neuropeptide Y. Nonetheless, the brain sites that send efferent SNS outflow to the bone have not yet been characterized. Using pseudorabies (PRV) viral transneuronal tracing, we report, for the first time, the identification of central SNS outflow sites that innervate bone. We find that the central SNS outflow to bone originates from 87 brain nuclei, sub-nuclei, and regions of six brain divisions, namely the midbrain and pons, hypothalamus, hindbrain medulla, fore-brain, cerebral cortex, and thalamus. We also find that certain sites, such as the raphe magnus (RMg) of the medulla and periaqueductal gray (PAG) of the midbrain, display greater degrees of PRV152 infection, suggesting that there is considerable site-specific variation in the levels of central SNS outflow to the bone. This comprehensive compendium illustrating the central coding and control of SNS efferent signals to bone should allow for a greater understanding of the neural regulation of bone metabolism, and importantly and of clinical relevance, mechanisms for central bone pain.

*For correspondence:
vitaly.ryu@mssm.edu (VR);
mone.zaidi@mountsinai.org (MZ)

## Editor's evaluation

This manuscript presents, for the first time, the utilization of PRV viral transneuronal tracing to elucidate the central coding and control mechanisms governing sympathetic nervous system (SNS) efferent signals to bone. This groundbreaking work not only holds promising research prospects but also establishes a robust foundation for understanding the neural regulation of bone metabolism.

## Introduction

Elegant studies have suggested that increased SNS tone causes bone loss through a reduction in bone formation, which is coupled with increased bone resorption (*Elefteriou, 2018*; *Elefteriou et al., 2005*; *Takeda et al., 2002*). It has also been shown using leptin-deficient mice with a high bone mass that the anti-osteogenic actions of leptin are mediated centrally by glucose-responsive neurons in the ventromedial hypothalamus through peripheral SNS pathways (*Ducy et al., 2000*; *Takeda et al., 2002*). Furthermore, both the periosteum and the bone marrow are innervated richly by the SNS as evidenced by immunoreactive tyrosine hydroxylase, dopamine beta-hydroxylase, or neuropeptide Y fibers. These latter SNS markers are associated mostly with the vasculature and SNS

vesicular acetylcholine transporter (VAChT), whereas vasoactive intestinal polypeptide (VIP) immunoreactive fibers display mainly a parenchymal location (*Francis et al., 1997*; *Hill and Elde, 1991*; *Hohmann et al., 1986*; *Martin et al., 2007*). Despite these studies, the distribution of SNS nerves within the mammalian skeleton and their connectivity to central neurons is far from being completely understood.

Viral transneuronal tracing has become an established technology to define central SNS outflow circuitry to peripheral organs. Bartha's K strain of the PRV is a transneuronal tract tracer that provides the ability to map multi-synaptic circuits within the same animal (*Ekstrand et al., 2008*; *Enquist, 2002*; *Song et al., 2005a*). Once in the host, PRVs are endocytosed at axon terminal membranes after binding to viral attachment proteins, which act as 'viral receptors.' Transported exclusively in a retrograde manner from the dendrites of the infected neurons to axons, PRVs first make synaptic contact with neuronal cell bodies, and undergo self-amplification and thereafter continue their specific backward ascent (*Curanovic and Enquist, 2009*). This results in an infection that progresses along the neuroaxis chain from the periphery to higher CNS sites (*Ekstrand et al., 2008*; *Enquist, 2002*; *Song et al., 2005a*).

Utilizing this viral technology, we have previously shown postganglionic SNS innervation of specific white and brown adipose tissue depots with the separate and shared central SNS relay sites (*Ryu and Bartness, 2014*; *Ryu et al., 2015*). Moreover, we have established a direct neuroanatomical connection between phosphodiesterase 5 A (PDE5A)-containing neurons in specific brain nuclei and bone, inferring a contribution of the central nodes to the bone-forming actions of PDE5A inhibitors (*Kim et al., 2020*). A hierarchical circuit controlling SNS output to rat femoral epiphyseal bone marrow has also been defined by identifying PRVs in the ganglia and paravertebral chain in the intermediolateral column (IML) of the lower thoracic spinal cord (*Dénes et al., 2005*). In addition, neurons in C1, A5, A7 catecholaminergic cell groups, several other nuclei of the ventrolateral and ventromedial medulla, periaqueductal gray, the paraventricular hypothalamic nucleus, among other hypothalamic nuclei, as well as the insular and piriform cortex comprise the known central network sending SNS outflow to bone marrow (*Dénes et al., 2005*). However, no studies have yet mapped the exact localization and organization of the central SNS circuitry innervating the murine femur. The purpose of the present study was thus to identify central SNS sites innervating bone and to investigate whether separate or/ and shared central SNS circuitries underpin the autonomic mediation of bone.

## Results
### Validation

Following PRV152 infections, mice remained asymptomatic until day 5 post-inoculation, after which time, mice began to display symptoms of infection, including occasional loss of body weight and decreased mobility, but most often an ungroomed coat. Mice were euthanized for histological analyses when such symptoms became apparent. Four of six mice were equally infected by PRV152 throughout the neuroaxis from the hindbrain to the forebrain and therefore were included in the analyses. Two mice exhibited over-infection by PRV152, as evidenced by widespread cloudy plaques surrounding the infected neurons; these mice were excluded from the analysis. We also found PRV152-labeled neurons in the IML of the spinal cord in accordance with our previous studies that defined SNS innervations of fat pads in the Siberian hamster (*Ryu and Bartness, 2014*; *Ryu et al., 2015*; *Ryu et al., 2017*).

Unilateral PRV152 microinjection into the right femur appeared bilaterally in the brain with almost unnoticeable domination of the viral infection between the two hemispheres. Likewise, prior studies on SNS and sensory innervations of various fat depots, utilizing the SNS tract tracer PRV152 and sensory system tract tracer HSV-1 produced no ipsilateral differences between the innervation patterns of SNS or sensory system with unilateral viral inoculation (*Bamshad et al., 1999*; *Leitner and Bartness, 2009*; *Song et al., 2008*; *Vaughan and Bartness, 2012*).

To validate the retrograde tract tracing methodology, we placed PRV152 at the same titer on the bone surface, rather than injecting it into the periosteum or metaphysis. No EGFP signal was detected in the PVH that is known to possess sympathetic pre-autonomic neurons or in the RPa (*Figure 1A*). By contrast, PRV152 injections into the periosteum or metaphysis resulted in positive EGFP immunostaining in the PVH (*Figure 1B*). In addition, we found PRV152-infected neurons in the IML of the spinal cord, at T13-L2 levels (*Figure 1B*), suggesting specific bone-SNS ganglia-IML-brain route of infection;

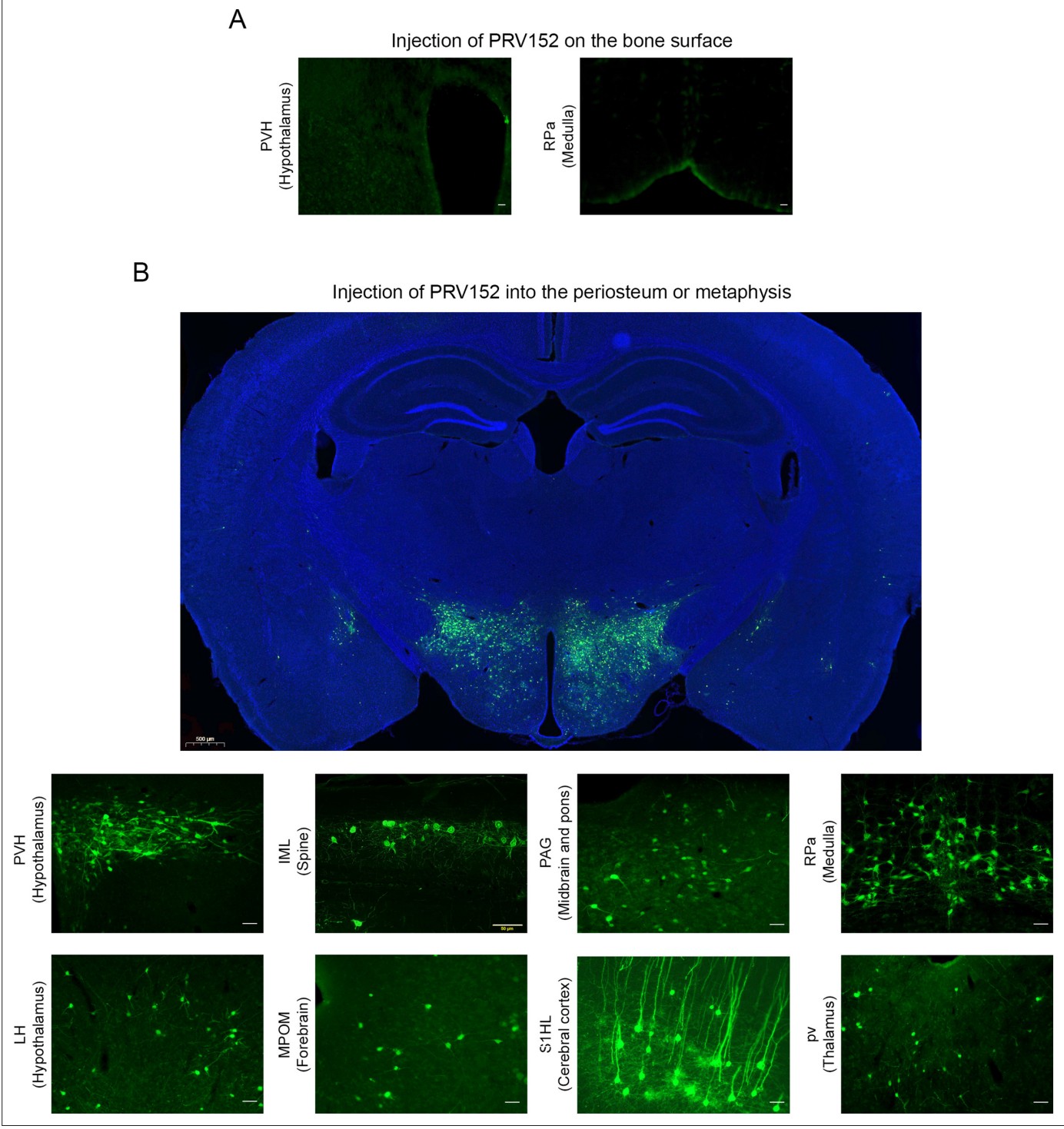

**Figure 1.** PRV152 transneuronal viral tract tracing. (**A**) As a control for viral injection, no expresses enhanced green fluorescent protein (EGFP) signal was detected in the paraventricular nucleus (PVH), known to possess main sympathetic pre-autonomic neurons, and the RPa, when PRV152 was placed on the bone surface. (**B**) By contrast, PRV152 injections into the periosteum or metaphyseal bone resulted in positive EGFP immunoreactivity in the PVH. In addition, we found PRV152-infected neurons in the intermediolateral cell column (IML) of the spinal cord at T13-L2 levels, suggesting specific bone-sympathetic nervous system (SNS) ganglia-IML-brain route of infection which are in concordance with our previous findings where PRV152 individually infected the classic SNS spinal cord neurons. Also shown are representative microphotographs illustrating PRV152 immunolabeling in the PAG (midbrain and pons), RPa (medulla), LH (hypothalamus), MPOM (forebrain), S1HL (cerebral cortex), and pv (thalamus). PVH, paraventricular hypothalamic nucleus; PAG, periaqueductal gray; RPa, raphe pallidus; LH, lateral hypothalamus; MPOM, medial preoptic nucleus, medial part; S1HL, primary somatosensory cortex, hindlimb region; pv, periventricular fiber system. Scale bar = 50 μm. Also shown is a representative low-magnification image at the hypothalamus neuroanatomical level (scale bar = 500 μm).

this is consistent with prior findings wherein PRV152 individually infected the classic SNS spinal cord neurons (*Bamshad et al., 1999*; *Ryu and Bartness, 2014*; *Ryu et al., 2015*).

## Viral infections in the brain

We identified 87 PRV152-positive brain nuclei, sub-nuclei, and regions within six brain divisions, with the hypothalamus having the most PRV152-infected SNS neurons connecting to bone (1177.25±62.75),

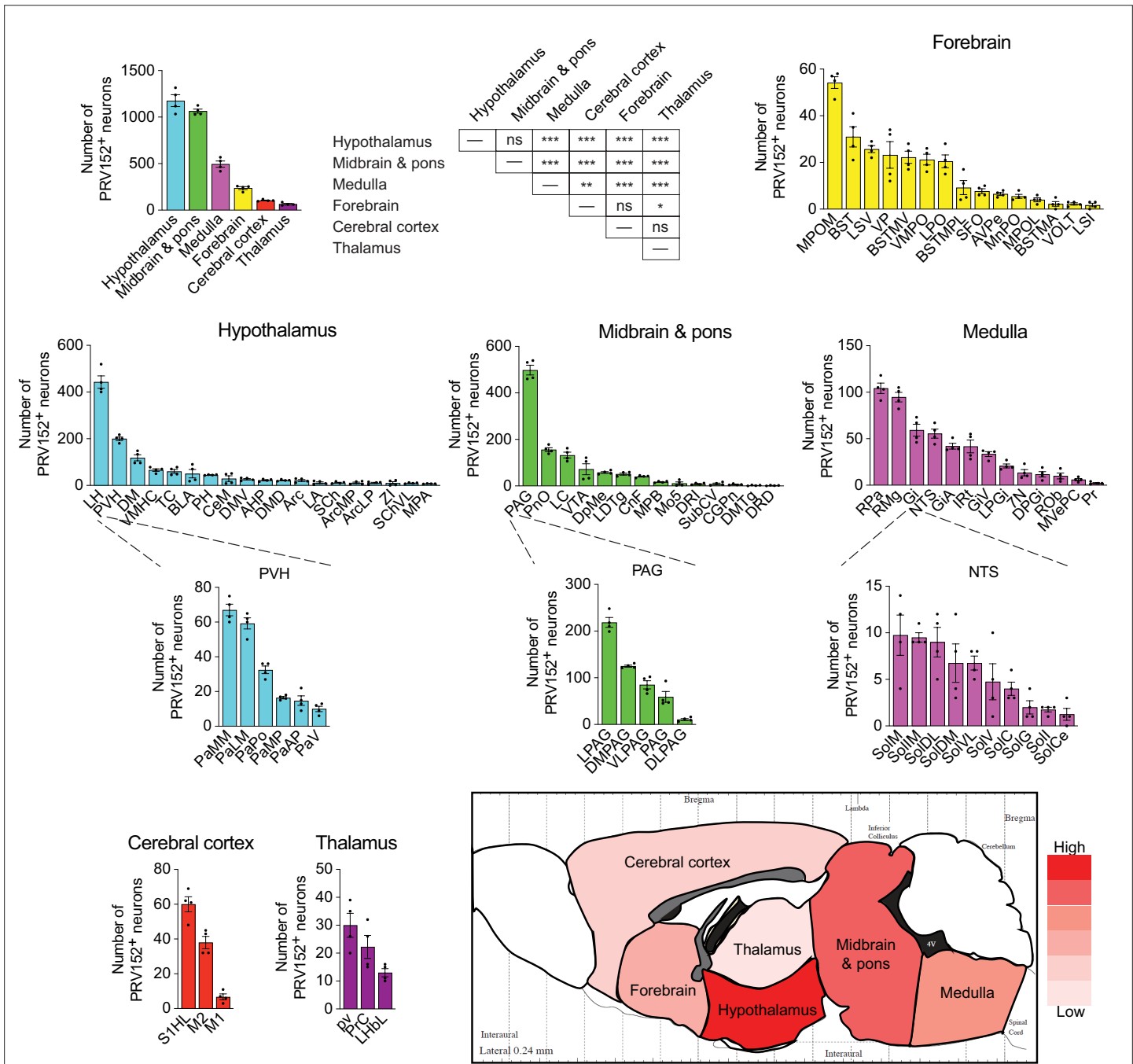

**Figure 2.** PRV152 immunolabeling in brain regions, sub-regions, and nuclei. Numbers, and heat map representation of PRV152-labeled neurons in brain regions, namely, hypothalamus, midbrain and pons, medulla, forebrain, cerebral cortex, and thalamus, as well as their sub-regions and nuclei, following viral injections into bone. n=4. Statistics: Mean ± SEM, two-tailed Student's *t*-test, *p<0.05, **p<0.01, ***p<0.001, ns (no significance).

The online version of this article includes the following source data for figure 2:

**Source data 1.** Contains the numerical data used to generate the figures.

followed, in descending order, by midbrain and pons (1065±22.39), hindbrain medulla (495.25±33.49), forebrain (237.5±15.08), cerebral cortex (104.75±4.64) and thalamus (65.25±7.78) (*Figure 2*). Hypothalamic areas with the highest percentages of PRV152-labeled neurons included the lateral hypothalamus (LH), PVH and dorsomedial hypothalamus (DM) (*Figure 1B* and *Figure 2*; see Appendix for a glossary of brain nuclei, sub-nuclei and regions). The LH and PVH also were among the regions with the highest absolute numbers of infected neurons of the 25 PRV152-positive nuclei, sub-nuclei, and regions. In the midbrain and pons, areas with the highest percentages and counts of PRV152-infected neurons included the PAG, lateral PAG (LPAG) and pontine reticular nucleus, oral part (PnO), among 18 nuclei, sub-nuclei, and regions. Single-labeled neurons were also notable in the hindbrain medulla, where the raphe pallidus nucleus (RPa), RMg, and gigantocellular reticular nucleus (Gi) were among 23 nuclei, sub-nuclei and regions, heavily represented by the largest percentages and counts of PRV152-labeled neurons. The forebrain areas with the highest percentages and numbers of PRV152-labeled neurons were the medial preoptic nucleus, medial part (MPOM), bed nucleus of the stria terminalis (BST), and lateral septal nucleus, ventral part (LSV) among 15 PRV152-positive nuclei, sub-nuclei, and regions. In the cerebral cortex, there were only three regions containing PRV152-labeled neurons–namely, the primary somatosensory cortex, hindlimb region (S1HL), and secondary and primary motor cortex (M2 and M1, respectively). The S1HL and M2 had both the highest percentages and numbers of PRV152-labeled neurons. Finally, we detected 3 brain sites with PRV152-infected neurons within the thalamus. Among the nuclei possessing the highest percentages and numbers of PRV152-labeled neurons were the periventricular fiber system (pv) and precommisural nucleus (PrC).

## Discussion

When injected into peripheral tissues, PRV152 travels exclusively in a retrograde manner *via* synaptically-linked neurons. First, the virus infects SNS postganglionic neurons at L1-L2 levels, followed by infection of SNS preganglionic neurons in the IML of the spinal cord and, finally, ascending to the brain. This yields a hierarchical chain of functionally connected neurons, thus allowing the mapping of the entire femur-brain neuroaxis. Using such transneuronal tract tracers, we (*Ryu and Bartness, 2014*; *Ryu et al., 2015*; *Ryu et al., 2017*) and others *Bamshad et al., 1998*; *Bowers et al., 2004*; *Shi and Bartness, 2001*; *Song and Bartness, 2001* have documented postganglionic SNS innervation of white and brown adipose tissue depots with the separate and shared central SNS nodes. Moreover, we have recently established a direct neuroanatomical link between PDE5A-containing neurons in specific brain sites and bone (*Kim et al., 2020*). We report here, for the first time, a comprehensive atlas that defines with remarkable precision the crosstalk between the SNS and bone. Notably, the PRV152 neural tract tracer especially predominates in the PAG of the midbrain, LH of the hypothalamus, RPa of the medulla, MPOM of the forebrain, S1HL of the cortex, and pv of the thalamus. Collectively, these data provide important insights into the distributed neural system integrating SNS neural circuitry with bone.

Neuroanatomical and functional evidence in mice suggests that the SNS regulates bone remodeling and bone mass (*Ducy et al., 2000*; *Francis et al., 1997*; *Hill and Elde, 1991*; *Hohmann et al., 1986*; *Martin et al., 2007*; *Takeda et al., 2002*). Furthermore, it is clear that leptin acts as an anti-osteogenesis signal through glucose responsive neurons in the VMH *via* peripheral SNS relay (*Takeda et al., 2002*). These data are consistent with histological evidence, using SNS markers in noradrenergic fibers, for a rich innervation of the periosteum and bone marrow (*Francis et al., 1997*; *Hill and Elde, 1991*; *Hohmann et al., 1986*; *Martin et al., 2007*). Likewise, dopamine-transporter-deficient mice with no rapid uptake of dopamine into presynaptic terminals are osteopenic (*Bliziotes et al., 2000*). Multisynaptic tract tracing has identified limited hierarchical central circuitry controlling SNS innervation of rat femoral epiphyseal bone marrow and bone (*Dénes et al., 2005*). Several SNS pathways from the brainstem and the hypothalamus relay to femoral bone marrow and the femur through preganglionic neurons in the lower thoracic and upper lumbar segments T4 to L1 of the IML and postganglionic neurons in paravertebral chain ganglia at lumbar levels (*Dénes et al., 2005*).

Despite the fact that largely the same brain sites project to both the femur (our findings) and bone marrow (*Dénes et al., 2005*), some sites display higher levels of PRV152 infectivity than others–this suggests that separate site-specific SNS circuits may project to the femur and femoral bone marrow. These overlapping SNS-innervating circuits to both sites include the midbrain PAG, somatosensory cortex, forebrain MPOM, thalamic periventricular nucleus, hypothalamic PVH, lateral hypothalamic

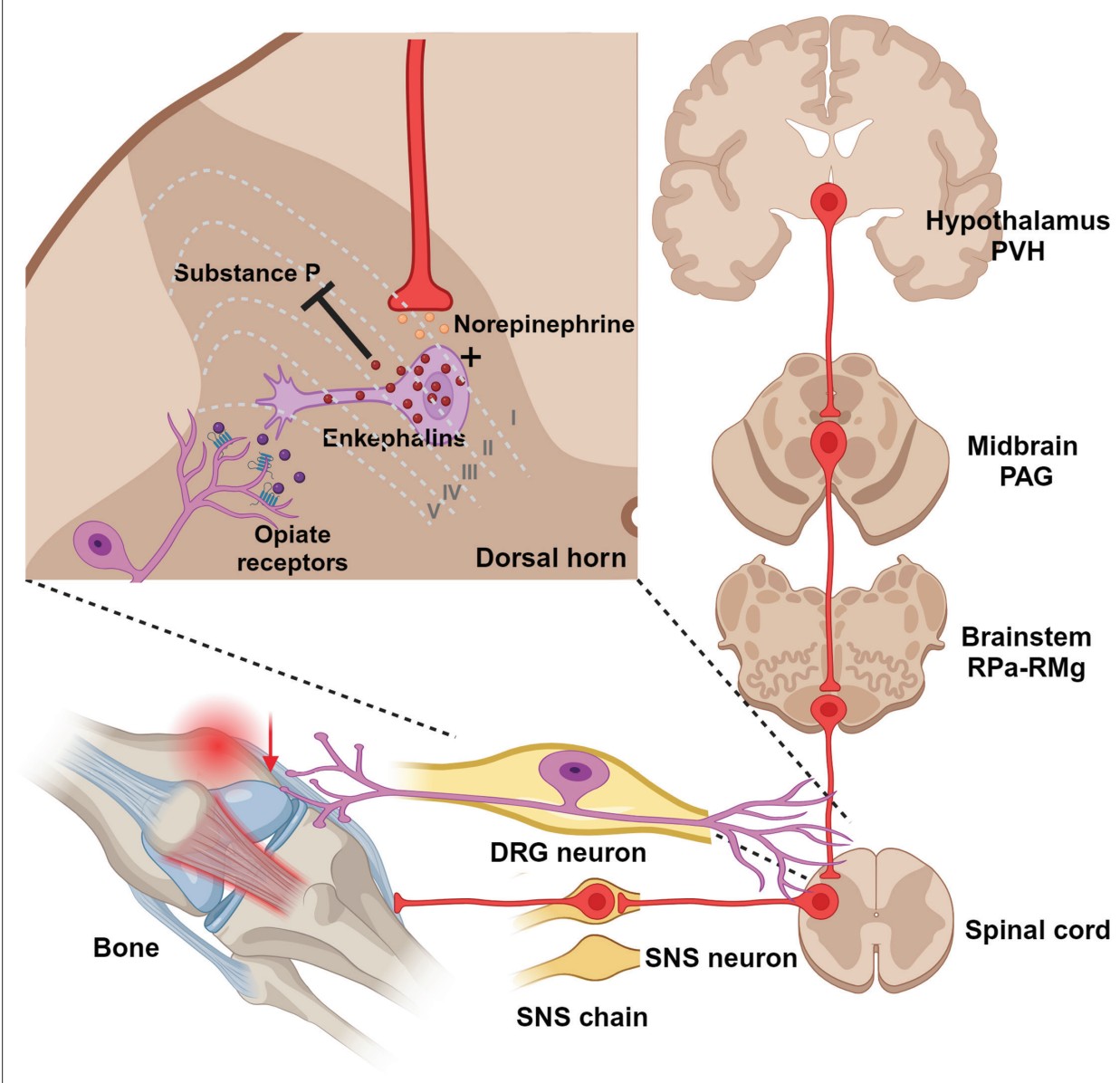

**Figure 3.** Diagrammatic outline of the sympathetic nervous system (SNS) brain–bone neuroaxis relevant to pain. The central SNS brain-bone circuit starts in the hypothalamic paraventricular nucleus (PVH) known to home SNS pre-autonomic neurons projecting to the SNS neurons of the periaqueductal gray (PAG) in the midbrain. From the PAG the SNS outflow is further relayed to the raphe pallidus-raphe magnus (RPa-RMg) neurons that are terminated in the dorsal horn of spinal gray matter, where they regulate the release of enkephalins that inhibit pain sensation by attenuating substance P (SP) release. In turn, opiates produce antinociception via the $\mu$-opiate receptors, in part, through modulation of responses to SP. Neurons in the RMg are involved in the central modulation of noxious stimuli, therefore, the RMg-PAG could be the part of the ascending hierarchical circuit relating to the perception of bone pain.

nucleus (LA), and medullary RPa. The PAG receives afferent fibers not only from the parabrachial nucleus and RPa (*Mantyh, 1982*), which contain PDE5A-expressing neurons sending SNS outputs to bone (*Kim et al., 2020*), but also from the spinal cord (*Pechura and Liu, 1986*). We and others have previously shown that the PAG sends SNS outflow to WAT in Siberian hamsters (*Bamshad et al., 1998*; *Nguyen et al., 2014*; *Ryu and Bartness, 2014*; *Song et al., 2005b*) and the laboratory rat (*Adler et al., 2012*). Most notably, the PAG is largely responsible for SNS responses and descending modulation of pain perception (*Baptista-de-Souza et al., 2018*; *Benarroch, 2008*; *Calvino and Grilo, 2006*). Therefore, this midbrain node could receive sensory inflow relating to bone pain and provide SNS relay (*Figure 3*).

Furthermore, nerves that innervate skeletal tissues seem to function in both skeletal pain and anabolism, affecting a variety of skeletal cell types. Identification of SNS neural nodes that regulate skeletal metabolism and nociceptive functions implies that manipulating the SNS or sensory neurons with drug therapies targeted to bone could have effects on bone pain and *vice versa*. Our ability to harness potential benefits for bone disease and pain alleviation *via* specific modulations in the SNS drive does not, therefore, seem remote.

We also find that two major nuclei in the hypothalamus–the PVH and LH–send SNS efferents to the bone. While the PVH, which is a home to major SNS pre-autonomic neurons, sends SNS projections to the bone marrow (*Dénes et al., 2005*), we find that LH predominantly innervates the femur. The functions of other hypothalamic SNS-bone feedback circuits are not presently known. However, given that the LH, PVH, and DM are the main brain regions that send SNS outflow to bone and also express leptin receptors (*Flak and Myers, 2016*), it is also possible that the anti-osteogenic relay for leptin might originate from neurons in the LH PVH, and/or DM.

Consistent with a prior study (*Dénes et al., 2005*), the highest number of PRV152-infected neurons in the medulla were the RPa and RMg. While we have previously established a contribution of PDE5A-containing neurons in the RPa to bone mass regulation (*Kim et al., 2020*), the functional role of the RMg in regulating bone remains unknown. Whereas projections from the raphe nuclei, including the RPa, terminate in the dorsal horn of spinal gray matter, where they regulate the release of enkephalins that inhibit pain sensation (*François et al., 2017*), RMg neurons are involved in the central modulation of noxious stimuli (*Fields et al., 1991*). Thus, the RMg-PAG could be the part of the ascending hierarchical circuit relating to the perception of bone pain. The importance of this circuit for the control of bone pain will require a more comprehensive demonstration of its pervasiveness across and within the mammalian species. Whether or not these findings can be extended to humans, they do provide important actionable targets for pain treatment.

In all, our results provide compelling evidence for a brain-bone SNS neuroaxis, likely part of coordinated and/or multiple redundant mechanisms that regulate bone metabolism and/or nociceptive functions. Furthermore, we show that bone is not innervated by unique neuron groups, but rather by overlapping SNS circuitry common to the control of other peripheral targets, such as bone marrow and adipose tissues. We believe our comprehensive atlas of the brain regions involved in coding and decoding SNS efferent signals to bone would stimulate further research into bone pain and the neural regulation of bone metabolism.

## Materials and methods

### Mice

Adult male mice (~3-4-month-old) were single-housed in a 12 hr:12 hr light:dark cycle at 22 ± 2 °C with ad libitum access to water and regular chow. All procedures were approved by the Mount Sinai Institutional Animal Care and Use Committee and were performed in accordance with Public Health Service and United States Department of Agriculture guidelines (IBC: SPROTO202200000224; IACUC: PROTO202100074).

### Viral injections

To identify brain sites sending the SNS outflow to bone, we used a transsynaptic tracing technique with a pseudorabies virus strain, PRV152. PRV152 expresses enhanced green fluorescent protein (EGFP) under control of the human cytomegalovirus immediate-early promoter. When injected into the femur, the virus travels exclusively in a retrograde manner *via* synaptically-linked neurons, first through SNS postganglionic neurons at L1-L2 levels, followed by infection of SNS preganglionic neurons in the IML of the spinal cord and, finally, ascending to the brain. This allows the mapping of the entire femur-brain neuroaxis.

All virus injections were performed according to Biosafety Level 2 standards. Mice (n=6) were anesthetized with isoflurane (2-3% in oxygen; Baxter Healthcare, Deerfield, IL) and the right femur-tibia joint was exposed for a series of PRV152 microinjections ($4.7 \times 10^9$ pfu/mL) into five loci (150 nL/locus) evenly distributed across the bone metaphysis and periosteum areas, which are known to be enriched with SNS innervation. The syringe was held in place for 60 s to prevent efflux of virus after each injection. Finally, the incision was closed with sterile sutures and wound clips. Nitrofurazone

powder (nfz Puffer; Hess & Clark, Lexington, KY) was applied locally to minimize the risk of bacterial infection. Note that, as a control for viral injection, we showed that no EGFP signal was detected when PRV152 was placed on the bone surface rather than injected into the periosteum or metaphyseal bone. In addition, we found PRV152-infected neurons in the IML of the spinal cords, suggesting specific bone-SNS ganglia-IML-brain route of infection, which is in concordance with our previous findings where PRV152 individually infected the classic SNS spinal cord neurons (*Bamshad et al., 1999*; *Ryu and Bartness, 2014*; *Ryu et al., 2015*). As a control for possible viral diffusion, we previously confirmed that the same virus titer and volume of PRV152 placed on the bone surface resulted in no infection in the sympathetic chain, spinal cord, and brain as opposed to intra-periosteum or intra-metaphyseal bone.

## Histology

Animals were sacrificed 6 days after the last PRV152 injection based on the progression of the virus to the brain in pilot studies (Ryu, V., unpublished observations). Mice were euthanized with carbon dioxide and perfused transcardially with 0.9% heparinized saline followed by 4% paraformaldehyde in 0.1 M phosphate-buffered saline (PBS; pH 7.4). Brains were collected and post-fixed in the same fixative for 3-4 hr at 4 °C, then transferred to a 30% sucrose solution in 0.1 M PBS with 0.1% sodium azide and stored at 4 °C until sectioning on a freezing stage sliding microtome at 25 μm. Sections were stored in 0.1 M PBS solution with 0.1% sodium azide until processing for immunofluorescence.

The EGFP reporter protein of the PRV152 is not resistant to fading once injected into peripheral tissues, therefore, a standard immunofluorescence procedure was utilized. For immunofluorescence, free-floating brain sections were rinsed in 0.1 M PBS (2×15 min) followed by a 30 min blocking in 10% normal goat serum (NGS; Vector Laboratories, Burlingame, CA) and 0.4% Triton X-100 in 0.1 M PBS. Next, sections were incubated with a primary chicken anti-EGFP antibody (1:1000; Thermo Fisher Scientific, catalog no. A10262) for 18 hr. Sections were then incubated in the secondary AlexaFluor-488-coupled goat anti-chicken antibody (1:700; Jackson Immunoresearch, catalog no. 103-545-155) with 2% NGS and 0.4% Triton X-100 in 0.1 M PBS at room temperature for 2 hr. For immunofluorescence controls, the primary antibody was either omitted or pre-adsorbed with the immunizing peptide overnight at 4 °C resulting in no immunoreactive staining. Sections were mounted onto slides (Superfrost Plus) and cover-slipped using ProLong Gold Antifade Reagent (Thermo Fisher Scientific, catalog no. P36982). All steps were performed at room temperature.

## Quantitation

Immunofluorescence images were viewed and captured using 10x and 20x magnification with an Observer.Z1 fluorescence microscope (Carl Zeiss, Germany) with appropriate filters for AlexaFluor-488 and DAPI. The single-labeled PRV152 and DAPI images were evaluated and overlaid using Zen software (Carl Zeiss, Germany) and ImageJ (NIH, Bethesda, MD). We counted cells positive for SNS PRV152 immunoreactivity in every sixth brain section using the manual tag feature of the Adobe Photoshop CS5.1 software, thus eliminating the likelihood of counting the same neurons more than once. Neuron numbers in the brain were averaged across each examined nucleus/sub-nucleus/region from all animals. A mouse brain atlas (*Paxinos and Franklin, 2007*) was used to identify brain areas. For the photomicrographs, we used Adobe Photoshop CS5.1 (Adobe Systems) only to adjust the brightness, contrast, and sharpness, to remove artifactual obstacles (i.e. obscuring bubbles) and to make the composite plates.

## Acknowledgements

Work at Icahn School of Medicine at Mount Sinai carried out at the Center for Translational Medicine and Pharmacology was supported by R01 AG071870 to MZ, TY, and S-MK; R01 AG074092 and U01 AG073148 to TY and MZ; and U19 AG060917 and R01 DK113627 to MZ.

## Additional information

### Competing interests

Vitaly Ryu, Daria Lizneva, Ki A Goosens, Se-Min Kim: Reviewing editor, *eLife*. Tony Yuen: Senior editor, *eLife*. Mone Zaidi: consults for Gershon Lehmann, Guidepoint and Coleman groups. The other authors declare that no competing interests exist.

### Funding

| Funder | Grant reference number | Author |
|---|---|---|
| National Institute on Aging | R01 AG071870 | Se-Min Kim<br>Tony Yuen<br>Mone Zaidi |
| National Institute on Aging | U01 AG073148 | Tony Yuen<br>Mone Zaidi |
| National Institute on Aging | R01 AG074092 | Tony Yuen<br>Mone Zaidi |
| National Institute on Aging | U19 AG060917 | Mone Zaidi |
| National Institute on Aging | R01 DK113627 | Mone Zaidi |
| National Institute of Diabetes and Digestive and Kidney Diseases | R01 DK107670 | Tony Yuen<br>Mone Zaidi |

The funders had no role in study design, data collection and interpretation, or the decision to submit the work for publication.

### Author contributions

Vitaly Ryu, Investigation, Methodology, Writing – original draft; Anisa Azatovna Gumerova, Liam Cullen, Data curation, Investigation; Ronit Witztum, Funda Korkmaz, Ofer Moldavski, Investigation; Hasni Kannangara, Data curation, Formal analysis; Orly Barak, Ki A Goosens, Supervision, Project administration; Daria Lizneva, Supervision, Investigation, Project administration; Sarah Stanley, Resources, Methodology; Se-Min Kim, Validation, Methodology; Tony Yuen, Formal analysis, Supervision, Funding acquisition, Writing – review and editing; Mone Zaidi, Conceptualization, Funding acquisition, Writing – original draft, Project administration, Writing – review and editing

### Author ORCIDs

Vitaly Ryu http://orcid.org/0000-0001-8068-4577
Ki A Goosens http://orcid.org/0000-0002-5246-2261
Mone Zaidi https://orcid.org/0000-0001-5911-9522

### Ethics

All procedures were approved by the Mount Sinai Institutional Animal Care and Use Committee and were performed in accordance with Public Health Service and United States Department of Agriculture guidelines (IBC: SPROTO202200000224; IACUC: PROTO202100074).

### Decision letter and Author response

Decision letter https://doi.org/10.7554/eLife.95727.sa1
Author response https://doi.org/10.7554/eLife.95727.sa2

## Additional files

### Supplementary files

• MDAR checklist

### Data availability

*Figure 2—source data 1* contains the numerical data used to generate the figures.

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

# Appendix 1

Glossary of the brain nuclei, sub-nuclei, and regions.

## Cerebral cortex

M1: primary motor cortex
M2: secondary motor cortex
S1HL: primary somatosensory cortex, hindlimb region

## Forebrain

AVPe: anteroventral periventricular nucleus
BST: bed nucleus of the stria terminalis
BSTMA: bed nucleus of the stria terminalis, medial division, anterior part
BSTMPL: bed nucleus of stria terminalis, medial division, posterolateral part
BSTMV: bed nucleus of the stria terminalis, medial division, ventral part
LPO: lateral preoptic area
LSI: lateral septal nucleus, intermediate part
LSV: lateral septal nucleus, ventral part
MnPO: median preoptic nucleus
MPOL: medial preoptic nucleus, lateral part
MPOM: medial preoptic nucleus, medial part
SFO: subfornical organ
VMPO: ventromedial preoptic nucleus
VOLT: vascular organ of the lamina terminalis
VP: ventral pallidum

## Thalamus

LhbL: lateral habenular nucleus, lateral part
PrC: precommissural nucleus
pv: periventricular fiber system

## Hypothalamus

AHP: anterior hypothalamic area, posterior part
Arc: arcuate hypothalamic nucleus
ArcLP: arcuate hypothalamic nucleus, lateroposterior part
ArcMP: arcuate hypothalamic nucleus, medial posterior part
BLA: basolateral amygdaloid nucleus, anterior part
CeM: central amygdaloid nucleus, medial division
DM: dorsomedial hypothalamic nucleus
DMD: dorsomedial hypothalamic nucleus, dorsal part
DMV: dorsomedial hypothalamic nucleus, ventral part
LA: lateroanterior hypothalamic nucleus
LH: lateral hypothalamic area
MPA: medial preoptic area
PaAP: paraventricular hypothalamic nucleus, anterior parvicellular part
PaLM: paraventricular hypothalamic nucleus, lateral magnocellular part
PaMM: paraventricular hypothalamic nucleus, medial magnocellular part
PaMP: paraventricular hypothalamic nucleus, medial parvicellular part
PaPo: paraventricular hypothalamic nucleus, posterior part

PaV: paraventricular hypothalamic nucleus, ventral part
PH: posterior hypothalamic area

