## [Editor Report]

This manuscript presents, for the first time, the utilization of PRV viral transneuronal tracing to elucidate the central coding and control mechanisms governing sympathetic nervous system (SNS) efferent signals to bone. This groundbreaking work not only holds promising research prospects but also establishes a robust foundation for understanding the neural regulation of bone metabolism.

---

## [Decision Letter]

**Decision letter after peer review:**

Thank you for submitting your article "An Atlas of Brain-Bone Sympathetic Neural Circuits" for consideration by *eLife*. Your article has been reviewed by 3 peer reviewers, and the evaluation has been overseen by a Reviewing Editor and Christopher Huang as the Senior Editor.

Essential revisions (for the authors):

This is a generally well-executed set of study, with appropriate data largely supporting the conclusions. To enhance the manuscript, it is recommended to address two key points:

1) Improve the resolution and quality of some of the data, as suggested by the reviewers, and

2) provide a more comprehensive discussion and explanation to enhance clarity and accuracy.

*Reviewer #1 (Recommendations for the authors):*

This manuscript presents, for the first time, the utilization of PRV viral transneuronal tracing to elucidate the central coding and control mechanisms governing sympathetic nervous system (SNS) efferent signals to bone. This groundbreaking work not only holds promising research prospects but also establishes a robust foundation for understanding the neural regulation of bone metabolism. I have a couple of comments before this paper gets published:

1. In Figure 1B, the panel depicting the intermediolateral nucleus (IML) features two scale bars.

2. The schematic illustration in Figure 3 lacks precision. The authors aim to convey the signal transmission from the rostral ventromedial medulla (RPa-RMg) to the dorsal horn of the spinal cord. However, the overall flowchart suggests involvement in the lateral and even ventral horn of the spinal cord. In the magnified dorsal horn schematic, it is advisable to meticulously delineate the specific laminae, considering the distinct laminar organization of the spinal cord.

3. Are there data available showcasing the outcomes post-injection of the control virus?

4. Are there low-magnification whole-brain images demonstrating the comprehensive distribution of green fluorescent protein in the mouse brain following retrograde tracing with the virus?

*Reviewer #2 (Recommendations for the authors):*

The study utilizes state-of the-art transsynaptic tracing technique with PRV152 virus that expresses enhanced green fluorescent protein to to identify the central sympathetic system outflow sites that innervate bone. Overall, the study is well designed and the manuscript is well written. The results in general are supportive of the conclusions.. There are a few issues that the authors may wish to address before publication.

1) For data shown in figure 2, it may be beneficial to perform statistical analyses to determine which of the regions show significantly higher number of PRV152+ neurons.

2) Since some of the signals secreted by sympathetic nerves (e.g. calcitonin gene related peptide, substance P, Sem3A) are known to influence metabolism, it is possible to identify whether any of the connections identified could be involved in regulating the activity of these neuropeptides?

3) It is stated that the animals exhibit symptoms of decreased mobility, body weight loss and infection after day 5 post-innoculation with PRV152. In this study, the animals were euthanized six days after the last PRV152 inoculation. Have the authors euthanized mice at day 4 or 5 before they start to develop symptoms to see if there are differences in brain-bone sympathetic neural circuits between symptomatic versus asymptomatic mice?

*Reviewer #3 (Recommendations for the authors):*

Authors used pseudorabies (PRV) viral transneuronal tracing and report for the first time, the identification of central SNS outflow sites that innervate bone. The authors found that the central SNS outflow to bone originates from brain nuclei, sub-nuclei and regions of six brain divisions (the midbrain and pons, hypothalamus, hindbrain medulla, forebrain, cerebral cortex, and thalamus). The authors provided compelling evidence for a brain-bone SNS neuroaxis that regulates bone metabolism and nociceptive functions. The paper is generally excellent with an interesting scientific premise and novel findings. Overall, the paper is strong and novel.

1) Authors may use arrows to point to infected neuron cells and indicate types of neurons in Figure 1.

2) Authors may discuss the mechanism by which how PRV152 migrates to brain after Injection of PRV152 into the periosteum or metaphysis.

3) Authors may explain why they did not directly detect EGFP expression on PRV152 infected cells under fluorescence microscope instead of using immunofluorescence approach to detect PRV152 infected cells.

4) Authors may discuss how their findings can benefit the research in the neural regulation of bone metabolism and nociceptive functions for bone pain.

5) Authors may provide a summary map of the Brain-Bone Sympathetic Neural Circuits derived from their findings in Figures 1 and 2. This map would enhance the manuscript's utility for readers.

6) Authors may discuss and summarize their results in detail for a better understanding of their findings

---

## [Author Response]

Essential revisions (for the authors):Reviewer #1 (Recommendations for the authors):This manuscript presents, for the first time, the utilization of PRV viral transneuronal tracing to elucidate the central coding and control mechanisms governing sympathetic nervous system (SNS) efferent signals to bone. This groundbreaking work not only holds promising research prospects but also establishes a robust foundation for understanding the neural regulation of bone metabolism. I have a couple of comments before this paper gets published:1. In Figure 1B, the panel depicting the intermediolateral nucleus (IML) features two scale bars.

Thank you. One of the scale bars has been removed.

2. The schematic illustration in Figure 3 lacks precision. The authors aim to convey the signal transmission from the rostral ventromedial medulla (RPa-RMg) to the dorsal horn of the spinal cord. However, the overall flowchart suggests involvement in the lateral and even ventral horn of the spinal cord. In the magnified dorsal horn schematic, it is advisable to meticulously delineate the specific laminae, considering the distinct laminar organization of the spinal cord.

We thank the reviewer for the comment. In the revised Figure 3, the mentioned errors have been fixed and the magnified dorsal horn schematic shows laminae.

3. Are there data available showcasing the outcomes post-injection of the control virus?

Regretfully, there is no control virus for PRV152. As a control for possible viral diffusion, we previously confirmed that the same virus titer and volume of PRV152 placed on the bone surface resulted in no infection in the sympathetic chain, spinal cord, and brain as opposed to intra-periosteum or intra-metaphyseal bone. We have added the latter sentence in the “Viral injections” part of the manuscript (page 6, lines 114-117).

4. Are there low-magnification whole-brain images demonstrating the comprehensive distribution of green fluorescent protein in the mouse brain following retrograde tracing with the virus?

We do have low-magnification whole-brain images demonstrating the comprehensive distribution of GFP in the brain following PRV152 injections. The representative image at the hypothalamus neuroanatomical level has been included as a Figure 1, low magnification.

Reviewer #2 (Recommendations for the authors):The study utilizes state-of the-art transsynaptic tracing technique with PRV152 virus that expresses enhanced green fluorescent protein to to identify the central sympathetic system outflow sites that innervate bone. Overall, the study is well designed and the manuscript is well written. The results in general are supportive of the conclusions.. There are a few issues that the authors may wish to address before publication.1) For data shown in figure 2, it may be beneficial to perform statistical analyses to determine which of the regions show significantly higher number of PRV152+ neurons.

In response to review comment, we have included a statistical comparison for the main brain regions.

2) Since some of the signals secreted by sympathetic nerves (e.g. calcitonin gene related peptide, substance P, Sem3A) are known to influence metabolism, it is possible to identify whether any of the connections identified could be involved in regulating the activity of these neuropeptides?

We thank the reviewer for the comment. Since CGRP and substance P are primarily released by the C and Aδ sensory nerves, it would be of great interest to identify brain centers receiving sensory inflow from bone using the sensory H129 virus in the future. This approach would shed more light into SNS-sensory regulation of CGRP and substance P. As with Sem3A, its role in the osteogenesis by altering the patterning of SNS innervations of bone remains unclear.

3) It is stated that the animals exhibit symptoms of decreased mobility, body weight loss and infection after day 5 post-innoculation with PRV152. In this study, the animals were euthanized six days after the last PRV152 inoculation. Have the authors euthanized mice at day 4 or 5 before they start to develop symptoms to see if there are differences in brain-bone sympathetic neural circuits between symptomatic versus asymptomatic mice?

We thank the reviewer for the comment. The first author performed a thorough time–course analysis of PRV152 viral progression through the SNS efferents in the Bartness lab at Georgia State University and found that optimal PRV152 progression and infection of the brain occur 6 days after PRV152 injections. If the mice are euthanized at earlier time points when mice are still asymptomatic, little or no PRV152 infection is detected in the brain.

Reviewer #3 (Recommendations for the authors):Authors used pseudorabies (PRV) viral transneuronal tracing and report for the first time, the identification of central SNS outflow sites that innervate bone. The authors found that the central SNS outflow to bone originates from brain nuclei, sub-nuclei and regions of six brain divisions (the midbrain and pons, hypothalamus, hindbrain medulla, forebrain, cerebral cortex, and thalamus). The authors provided compelling evidence for a brain-bone SNS neuroaxis that regulates bone metabolism and nociceptive functions. The paper is generally excellent with an interesting scientific premise and novel findings. Overall, the paper is strong and novel.1) Authors may use arrows to point to infected neuron cells and indicate types of neurons in Figure 1.

We thank the reviewer for the comment. All EGFP-positive neurons are SNS neurons infected with PRV152. In this study, we did not use neuronal markers.

2) Authors may discuss the mechanism by which how PRV152 migrates to brain after Injection of PRV152 into the periosteum or metaphysis.

We thank the reviewer for the great comment and added the part explaining how PRV152 migrates to the brain from the sites of inoculation in femur (page 6, lines 97-100 and page 10, lines 212-216).

3) Authors may explain why they did not directly detect EGFP expression on PRV152 infected cells under fluorescence microscope instead of using immunofluorescence approach to detect PRV152 infected cells.

Thank you for the comment. We’ve added an explanation on why we conducted immunofluorescence (page 7, lines 127-128).

4) Authors may discuss how their findings can benefit the research in the neural regulation of bone metabolism and nociceptive functions for bone pain.

We fully concur with the review comment and further discussed how our findings can benefit bone metabolism and pain (page 12, lines 256-261).

5) Authors may provide a summary map of the Brain-Bone Sympathetic Neural Circuits derived from their findings in Figures 1 and 2. This map would enhance the manuscript's utility for readers.

Great suggestion and we added a summary map in Figure 2.

6) Authors may discuss and summarize their results in detail for a better understanding of their findings

We thank the reviewer for this comment and have addressed the concern in the discussion (page 10, lines 211-216 and page 12, lines 256-261).